# Probabilistic Models for Integration Error in the Assessment of Functional Cardiac Models

**Chris. J. Oates**[1,5]**, Steven Niederer**[2]**, Angela Lee**[2]**, François-Xavier Briol**[3]**, Mark Girolami**[4,5]
[1]Newcastle University, [2]King's College London, [3]University of Warwick,
[4]Imperial College London, [5]Alan Turing Institute

## Abstract

This paper studies the numerical computation of integrals, representing estimates or predictions, over the output $f(x)$ of a computational model with respect to a distribution $p(\mathrm{d}x)$ over uncertain inputs $x$ to the model. For the functional cardiac models that motivate this work, neither $f$ nor $p$ possess a closed-form expression and evaluation of either requires $\approx 100$ CPU hours, precluding standard numerical integration methods. Our proposal is to treat integration as an estimation problem, with a joint model for both the a priori unknown function $f$ and the a priori unknown distribution $p$. The result is a posterior distribution over the integral that explicitly accounts for dual sources of numerical approximation error due to a severely limited computational budget. This construction is applied to account, in a statistically principled manner, for the impact of numerical errors that (at present) are confounding factors in functional cardiac model assessment.

## 1 Motivation: Predictive Assessment of Computer Models

This paper considers the problem of assessment for computer models [7], motivated by an urgent need to assess the performance of sophisticated functional cardiac models [25]. In concrete terms, the problem that we consider can be expressed as the numerical approximation of integrals

$$p(f) \quad = \quad \int f(x)p(\mathrm{d}x), \tag{1}$$

where $f(x)$ denotes a functional of the output from a computer model and $x$ denotes unknown inputs (or 'parameters') of the model. The term $p(x)$ denotes a posterior distribution over model inputs. Although not our focus in this paper, we note that $p(x)$ is defined based on a prior $\pi_0(x)$ over these inputs and training data $y$ assumed to follow the computer model $\pi(y|x)$ itself. The integral $p(f)$, in our context, represents a posterior prediction of actual cardiac behaviour. The computational model can be assessed through comparison of these predictions to test data generated from an experiment.

The challenging nature of cardiac models – and indeed computer models in general – is such that a closed-form for both $f(x)$ and $p(\mathrm{d}x)$ is precluded [23]. Instead, it is typical to be provided with a finite collection of samples $\{x_i\}_{i=1}^n$ obtained from $p(\mathrm{d}x)$ through Monte Carlo (or related) methods [32]. The integrand $f(x)$ is then evaluated at these $n$ input configurations, to obtain $\{f(x_i)\}_{i=1}^n$. Limited computational budgets necessitate that the number $n$ is small and, in such situations, the error of an estimator for the integral $p(f)$ based on the data $\{(x_i, f(x_i))\}_{i=1}^n$ is subject to strict information-theoretic lower bounds [26]. The practical consequence is that an unknown (non-negligible) numerical error is introduced in the numerical approximation of $p(f)$, unrelated to the performance of the model. If this numerical error is ignored, it will constitute a confounding factor in the assessment of predictive performance for the computer model. It is therefore unclear how a fair model assessment can proceed. This motivates an attempt to understand the extent of numerical error in any estimate of $p(f)$. This is non-trivial; for example, the error distribution of the arithmetic mean $\frac{1}{n}\Sigma_{i=1}^n f(x_i)$ depends on the

unknown $f$ and $p$, and attempts to estimate this distribution solely from data, e.g. via a bootstrap or a central limit approximation, *cannot succeed* in general when the number of samples $n$ is small [27].

Our first contribution, in this paper, is to argue that approximation of $p(f)$ from samples $\{x_i\}_{i=1}^n$ and function evaluations $\{f(x_i)\}_{i=1}^n$ can be cast as an estimation task. Our second contribution is to derive a posterior distribution over the unknown value $p(f)$ of the integral. This distribution provides an interpretable quantification of the extent of numerical integration error that can be reasoned with and propagated through subsequent model assessment. Our third contribution is to establish theoretical properties of the proposed method. The method we present falls within the framework of *Probabilistic Numerics* and our work can be seen as a contribution to this emerging area [16, 5]. In particular, the method proposed is reminiscent of *Bayesian Quadrature* (BQ) [9, 28, 29, 15]. In BQ, a Gaussian prior measure is placed on the unknown function $f$ and is updated to a posterior when conditioned on the information $\{(x_i, f(x_i))\}_{i=1}^n$. This induces both a prior and a posterior over the value of $p(f)$ as push-forward measures under the projection operator $f \mapsto p(f)$. Since its introduction, several authors have related BQ to other methods such as the 'herding' approach from machine learning [17, 3], random feature approximations used in kernel methods [1], classical quadrature rules [33] and Quasi Monte Carlo (QMC) methods [4]. Most recently, [21] extended theoretical results for BQ to misspecified prior models, and [22] who provided efficient matrix algebraic methods for the implementation of BQ. However, as an important point of distinction, notice that BQ pre-supposes $p(\mathrm{d}x)$ is known in closed-form - it does not apply in situations where $p(\mathrm{d}x)$ is instead sampled. In this latter case $p(\mathrm{d}x)$ will be called an *intractable* distribution and, for model assessment, this scenario is typical.

To extend BQ to intractable distributions, this paper proposes to use a Dirichlet process mixture prior to estimate the unknown distribution $p(\mathrm{d}x)$ from Monte Carlo samples $\{x_i\}_{i=1}^n$ [12]. It will be demonstrated that this leads to a simple expression for the closed-form terms which are required to implement the usual BQ. The overall method, called *Dirichlet process mixture Bayesian quadrature* (DPMBQ), constructs a (univariate) distribution over the unknown integral $p(f)$ that can be exploited to tease apart the intrinsic performance of a model from numerical integration error in model assessment. Note that BQ was used to estimate marginal likelihood in e.g. [30]. The present problem is distinct, in that we focus on predictive performance (of posterior expectations) rather than marginal likelihood, and its solution demands a correspondingly different methodological development.

On the computational front, DPMBQ costs $O(n^3)$. However, this cost is de-coupled from the often orders-of-magnitude larger costs involved in both evaluation of $f(x)$ and $p(\mathrm{d}x)$, which form the main computational bottleneck. Indeed, in the modern computational cardiac models that motivate this research, the $\approx 100$ CPU hour time required for a single simulation limits the number $n$ of available samples to $\approx 10^3$ [25]. At this scale, numerical integration error cannot be neglected in model assessment. This raises challenges when making assessments or comparisons between models, since the intrinsic performance of models cannot be separated from numerical error that is introduced into the assessment. Moreover, there is an urgent ethical imperative that the clinical translation of such models is accompanied with a detailed quantification of the unknown numerical error component in model assessment. Our contribution explicitly demonstrates how this might be achieved.

The remainder of the paper proceeds as follows: In Section 2.1 we first recall the usual BQ method, then in Section 2.2 we present and analyse our novel DPMBQ method. Proofs of theoretical results are contained in the electronic supplement. Empirical results are presented in Section 3 and the paper concludes with a discussion in Section 4.

## 2 Probabilistic Models for Numerical Integration Error

Consider a domain $\Omega \subseteq \mathbb{R}^d$, together with a distribution $p(\mathrm{d}x)$ on $\Omega$. As in Eqn. 1, $p(f)$ will be used to denote the integral of the argument $f$ with respect to the distribution $p(\mathrm{d}x)$. All integrands are assumed to be (measurable) functions $f : \Omega \to \mathbb{R}$ such that the integral $p(f)$ is well-defined. To begin, we recall details for the BQ method when $p(\mathrm{d}x)$ is known in closed-form [9, 28]:

### 2.1 Probabilistic Integration for Tractable Distributions (BQ)

In standard BQ [9, 28], a Gaussian Process (GP) prior $f \sim \mathrm{GP}(m, k)$ is assigned to the integrand $f$, with mean function $m : \Omega \to \mathbb{R}$ and covariance function $k : \Omega \times \Omega \to \mathbb{R}$ [see 31, for further details

on GPs]. The implied prior over the integral $p(f)$ is then the push-forward of the GP prior through the projection $f \mapsto p(f)$:

$$p(f) \sim \mathrm{N}(p(m), p \otimes p(k))$$

where $p \otimes p : \Omega \times \Omega \to \mathbb{R}$ is the measure formed by independent products of $p(\mathrm{d}x)$ and $p(\mathrm{d}x')$, so that under our notational convention the so-called *initial error* $p \otimes p(k)$ is equal to $\iint k(x, x') p(\mathrm{d}x) p(\mathrm{d}x')$. Next, the GP is conditioned on the information in $\{(x_i, f(x_i))\}_{i=1}^n$. The conditional GP takes a conjugate form $f | X, f(X) \sim \mathrm{GP}(m_n, k_n)$, where we have written $X = (x_1, \ldots, x_n)$, $f(X) = (f(x_1), \ldots, f(x_n))^\top$. Formulae for the mean function $m_n : \Omega \to \mathbb{R}$ and covariance function $k_n : \Omega \times \Omega \to \mathbb{R}$ are standard can be found in [31, Eqns. 2.23, 2.24]. The BQ posterior over $p(f)$ is the push forward of the GP posterior:

$$p(f) \mid X, f(X) \sim \mathrm{N}(p(m_n), p \otimes p(k_n)) \tag{2}$$

Formulae for $p(m_n)$ and $p \otimes p(k_n)$ were derived in [28]:

$$p(m_n) = f(X)^\top k(X, X)^{-1} \mu(X) \tag{3}$$

$$p \otimes p(k_n) = p \otimes p(k) - \mu(X)^\top k(X, X)^{-1} \mu(X) \tag{4}$$

where $k(X, X)$ is the $n \times n$ matrix with $(i, j)$th entry $k(x_i, x_j)$ and $\mu(X)$ is the $n \times 1$ vector with $i$th entry $\mu(x_i)$ where the function $\mu$ is called the *kernel mean* or *kernel embedding* [see e.g. 35]:

$$\mu(x) = \int k(x, x') p(\mathrm{d}x') \tag{5}$$

Computation of the kernel mean and the initial error each requires that $p(\mathrm{d}x)$ is known in general. The posterior in Eqn. 2 was studied in [4], where rates of posterior contraction were established under further assumptions on the smoothness of the covariance function $k$ and the smoothness of the integrand. Note that the matrix inverse of $k(X, X)$ incurs a (naive) computational cost of $O(n^3)$; however this cost is *post-hoc* and decoupled from (more expensive) computation that involves the computer model. Sparse or approximate GP methods could also be used.

## 2.2 Probabilistic Integration for Intractable Distributions

The dependence of Eqns. 3 and 4 on both the kernel mean and the initial error means that BQ cannot be used for intractable $p(\mathrm{d}x)$ in general. To address this we construct a second non-parametric model for the unknown $p(\mathrm{d}x)$, presented next.

**Dirichlet Process Mixture Model**  Consider an infinite mixture model

$$p(\mathrm{d}x) = \int \psi(\mathrm{d}x; \phi) P(\mathrm{d}\phi), \tag{6}$$

where $\psi : \Omega \times \Phi \to [0, \infty)$ is such that $\psi(\cdot; \phi)$ is a distribution on $\Omega$ with parameter $\phi \in \Phi$ and $P$ is a mixing distribution defined on $\Phi$. In this paper, each data point $x_i$ is modelled as an independent draw from $p(\mathrm{d}x)$ and is associated with a latent variable $\phi_i \in \Phi$ according to the generative process of Eqn. 6. i.e. $x_i \sim \psi(\cdot; \phi_i)$. To limit scope, the extension to correlated $x_i$ is reserved for future work.

The Dirichlet process (DP) is the natural conjugate prior for non-parametric discrete distributions [12]. Here we endow $P(\mathrm{d}\phi)$ with a DP prior $P \sim \mathrm{DP}(\alpha, P_b)$, where $\alpha > 0$ is a concentration parameter and $P_b(\mathrm{d}\phi)$ is a base distribution over $\Phi$. The base distribution $P_b$ coincides with the prior expectation $\mathbb{E}[P(\mathrm{d}\phi)] = P_b(\mathrm{d}\phi)$, while $\alpha$ determines the spread of the prior about $P_b$. The DP is characterised by the property that, for any finite partition $\Phi = \Phi_1 \cup \cdots \cup \Phi_m$, it holds that $(P(\Phi_1), \ldots, P(\Phi_m)) \sim \mathrm{Dir}(\alpha P_b(\Phi_1), \ldots, \alpha P_b(\Phi_m))$ where $P(S)$ denotes the measure of the set $S \subseteq \Phi$. For $\alpha \to 0$, the DP is supported on the set of atomic distributions, while for $\alpha \to \infty$, the DP converges to an atom on the base distribution. This overall approach is called a DP *mixture* (DPM) model [13].

For a random variable $Z$, the notation $[Z]$ will be used as shorthand to denote the density function of $Z$. It will be helpful to note that for $\phi_i \sim P$ independent, writing $\phi_{1:n} = (\phi_1, \ldots, \phi_n)$, standard conjugate results for DPs lead to the conditional

$$P \mid \phi_{1:n} \sim \mathrm{DP}\left(\alpha + n, \frac{\alpha}{\alpha + n} P_b + \frac{1}{\alpha + n} \sum_{i=1}^n \delta_{\phi_i}\right)$$

where $\delta_{\phi_i}(\mathrm{d}\phi)$ is an atomic distribution centred at the location $\phi_i$ of the $i$th sample in $\phi_{1:n}$. In turn, this induces a conditional $[\mathrm{d}p|\phi_{1:n}]$ for the unknown distribution $p(\mathrm{d}x)$ through Eqn. 6.

**Kernel Means via Stick Breaking** The *stick breaking* characterisation can be used to draw from the conditional DP [34]. A generic draw from $[P|\phi_{1:n}]$ can be characterised as

$$P(\mathrm{d}\phi) = \sum_{j=1}^{\infty} w_j \delta_{\varphi_j}(\mathrm{d}\phi), \qquad w_j = \beta_j \prod_{j'=1}^{j-1}(1-\beta_{j'}) \tag{7}$$

where randomness enters through the $\varphi_j$ and $\beta_j$ as follows:

$$\varphi_j \overset{\text{iid}}{\sim} \frac{\alpha}{\alpha+n}P_b + \frac{1}{\alpha+n}\sum_{i=1}^{n}\delta_{\phi_i}, \qquad \beta_j \overset{\text{iid}}{\sim} \text{Beta}(1,\alpha+n)$$

In practice the sum in Eqn. 7 may be truncated at a large finite number of terms, $N$, with negligible truncation error, since weights $w_j$ vanish at a geometric rate [18]. The truncated DP has been shown to provide accurate approximation of integrals with respect to the original DP [19]. For a realisation $P(\mathrm{d}\phi)$ from Eqn. 7, observe that the induced distribution $p(\mathrm{d}x)$ over $\Omega$ is

$$p(\mathrm{d}x) = \sum_{j=1}^{\infty} w_j \psi(\mathrm{d}x; \varphi_j). \tag{8}$$

Thus we have an alternative characterisation of $[p|\phi_{1:n}]$.

Our key insight is that one can take $\psi$ and $k$ to be a conjugate pair, such that both the kernel mean $\mu(x)$ and the initial error $p \otimes p(k)$ will be available in an explicit form for the distribution in Eqn. 8 [see Table 1 in 4, for a list of conjugate pairs]. For instance, in the one-dimensional case, consider $\varphi = (\varphi_1, \varphi_2)$ and $\psi(\mathrm{d}x; \varphi) = \text{N}(\mathrm{d}x; \varphi_1, \varphi_2)$ for some location and scale parameters $\varphi_1$ and $\varphi_2$. Then for the Gaussian kernel $k(x, x') = \zeta \exp(-(x-x')^2/2\lambda^2)$, the kernel mean becomes

$$\mu(x) = \sum_{j=1}^{\infty} \frac{\zeta \lambda w_j}{(\lambda^2 + \varphi_{j,2})^{1/2}} \exp\left(-\frac{(x-\varphi_{j,1})^2}{2(\lambda^2+\varphi_{j,2})}\right) \tag{9}$$

and the initial variance can be expressed as

$$p \otimes p(k) = \sum_{j=1}^{\infty}\sum_{j'=1}^{\infty} \frac{\zeta \lambda w_j w_{j'}}{(\lambda^2 + \varphi_{j,2} + \varphi_{j',2})^{1/2}} \exp\left(-\frac{(\varphi_{j,1}-\varphi_{j',1})^2}{2(\lambda^2+\varphi_{j,2}+\varphi_{j',2})}\right). \tag{10}$$

Similar calculations for the multi-dimensional case are straight-forward and provided in the Supplemental Information.

**The Proposed Model** To put this all together, let $\theta$ denote all hyper-parameters that (a) define the GP prior mean and covariance function, denoted $m_\theta$ and $k_\theta$ below, and (b) define the DP prior, such as $\alpha$ and the base distribution $P_b$. It is assumed that $\theta \in \Theta$ for some specified set $\Theta$. The marginal posterior distribution for $p(f)$ in the DPMBQ model is defined as

$$[p(f) \mid X, f(X)] = \iint [p(f) \mid X, f(X), p, \theta]\,[\mathrm{d}p \mid X, \theta]\,[\mathrm{d}\theta]. \tag{11}$$

The first term in the integral is BQ for a fixed distribution $p(\mathrm{d}x)$. The second term represents the DPM model for the unknown $p(\mathrm{d}x)$, while the third term $[\mathrm{d}\theta]$ represents a hyper-prior distribution over $\theta \in \Theta$. The DPMBQ distribution in Eqn. 11 does not admit a closed-form expression. However, it is straight-forward to sample from this distribution without recourse to $f(x)$ or $p(\mathrm{d}x)$. In particular, the second term can be accessed through the law of total probabilities:

$$[\mathrm{d}p \mid X, \theta] = \int [\mathrm{d}p \mid \phi_{1:n}]\,[\phi_{1:n} \mid X, \theta]\,\mathrm{d}\phi_{1:n}$$

where the first term $[\mathrm{d}p \mid \phi_{1:n}]$ is the stick-breaking construction and the term $[\phi_{1:n} \mid X, \theta]$ can be targeted with a Gibbs sampler. Full details of the procedure we used to sample from Eqn. 11, which is de-coupled from the much larger costs associated with the computer model, are provided in the Supplemental Information.

**Theoretical Analysis** The analysis reported below restricts attention to a fixed hyper-parameter $\theta$ and a one-dimensional state-space $\Omega = \mathbb{R}$. The extension of theoretical results to multiple dimensions was beyond the scope of this paper.

Our aim in this section is to establish when DPMBQ is "consistent". To be precise, a random distribution $\mathbb{P}_n$ over an unknown parameter $\zeta \in \mathbb{R}$, whose true value is $\zeta_0$, is called *consistent* for $\zeta_0$ at a *rate* $r_n$ if, for all $\delta > 0$, we have $\mathbb{P}_n[(-\infty, \zeta_0 - \delta) \cup (\zeta_0 + \delta, \infty)] = O_P(r_n)$. Below we denote with $f_0$ and $p_0$ the respective true values of $f$ and $p$; our aim is to estimate $\zeta_0 = p_0(f_0)$. Denote with $\mathcal{H}$ the reproducing kernel Hilbert space whose reproducing kernel is $k$ and assume that the GP prior mean $m$ is an element of $\mathcal{H}$. Our main theoretical result below establishes that the DPMBQ posterior distribution in Eqn. 11, which is a random object due to the $n$ independent draws $x_i \sim p(\mathrm{d}x)$, is consistent:

**Theorem.** *Let $P_0$ denote the true mixing distribution. Suppose that:*

1. *$f$ belongs to $\mathcal{H}$ and $k$ is bounded on $\Omega \times \Omega$.*
2. *$\psi(\mathrm{d}x; \varphi) = \mathrm{N}(\mathrm{d}x; \varphi_1, \varphi_2)$.*
3. *$P_0$ has compact support $\mathrm{supp}(P_0) \subset \mathbb{R} \times (\underline{\sigma}, \overline{\sigma})$ for some fixed $\underline{\sigma}, \overline{\sigma} \in (0, \infty)$.*
4. *$P_b$ has positive, continuous density on a rectangle $R$, s.t. $\mathrm{supp}(P_b) \subseteq R \subseteq \mathbb{R} \times [\underline{\sigma}, \overline{\sigma}]$.*
5. *$P_b(\{(\varphi_1, \varphi_2) : |\varphi_1| > t\}) \leq c \exp(-\gamma |t|^{\delta})$ for some $\gamma, \delta > 0$ and $\forall \, t > 0$.*

*Then the posterior $\mathbb{P}_n = [p(f) \mid X, f_0(X)]$ is consistent for the true value $p_0(f_0)$ of the integral at the rate $n^{-1/4+\epsilon}$ where the constant $\epsilon > 0$ can be arbitrarily small.*

The proof is provided in the Supplemental Information. Assumption (1) derives from results on consistent BQ [4] and can be relaxed further with the results in [21] (not discussed here), while assumptions (2-5) derive from previous work on consistent estimation with DPM priors [14]. For the case of BQ when $p(\mathrm{d}x)$ is known and $\mathcal{H}$ a Sobolev space of order $s > 1/2$ on $\Omega = [0,1]$, the corresponding posterior contraction rate is $\exp(-Cn^{2s-\epsilon})$ [4, Thm. 1]. Our work, while providing only an upper bound on the convergence rate, suggests that there is an increase in the fundamental complexity of estimation for $p(\mathrm{d}x)$ unknown compared to $p(\mathrm{d}x)$ known. Interestingly, the $n^{-1/4+\epsilon}$ rate is slower than the classical Bernstein-von Mises rate $n^{-1/2}$ [36]. However, an out-of-hand comparison between these two quantities is not straight forward, as the former involves the interaction of two distinct non-parametric statistical models. It is known Bernstein-von Mises results can be delicate for non-parametric problems [see, for example, the counter-examples in 10]. Rather, this theoretical analysis guarantees consistent estimation in a regime that is non-standard.

## 3  Results

The remainder of the paper reports empirical results from application of DPMBQ to simulated data and to computational cardiac models.

### 3.1  Simulation Experiments

To explore the empirical performance of DPMBQ, a series of detailed simulation experiments were performed. For this purpose, a flexible test bed was constructed wherein the true distribution $p_0$ was a normal mixture model (able to approximate any continuous density) and the true integrand $f_0$ was a polynomial (able to approximate any continuous function). In this set-up it is possible to obtain closed-form expressions for all integrals $p_0(f_0)$ and these served as a gold-standard benchmark. To mimic the scenario of interest, a small number $n$ of samples $x_i$ were drawn from $p_0(\mathrm{d}x)$ and the integrand values $f_0(x_i)$ were obtained. This information $X$, $f_0(X)$ was provided to DPMBQ and the output of DPMBQ, a distribution over $p(f)$, was compared against the actual value $p_0(f_0)$ of the integral. For all experiments in this paper the Gaussian kernel $k$ defined in Sec. 2.2 was used; the integrand $f$ was normalised and the associated amplitude hyper-parameter $\zeta = 1$ fixed, whereas the length-scale hyper-parameter $\lambda$ was assigned a $\mathrm{Gam}(2,1)$ hyper-prior. For the DPM, the concentration parameter $\alpha$ was assigned a $\mathrm{Exp}(1)$ hyper-prior. These choices allowed for adaptation of DPMBQ to the smoothness of both $f$ and $p$ in accordance with the data presented to the method. The base distribution $P_b$ for DPMBQ was taken to be normal inverse-gamma with hyper-parameters $\mu_0 = 0$, $\lambda_0 = \alpha_0 = \beta_0 = 1$, selected to facilitate a simplified Gibbs sampler. Full details of the simulation set-up and Gibbs sampler are reported in the Supplemental Information.

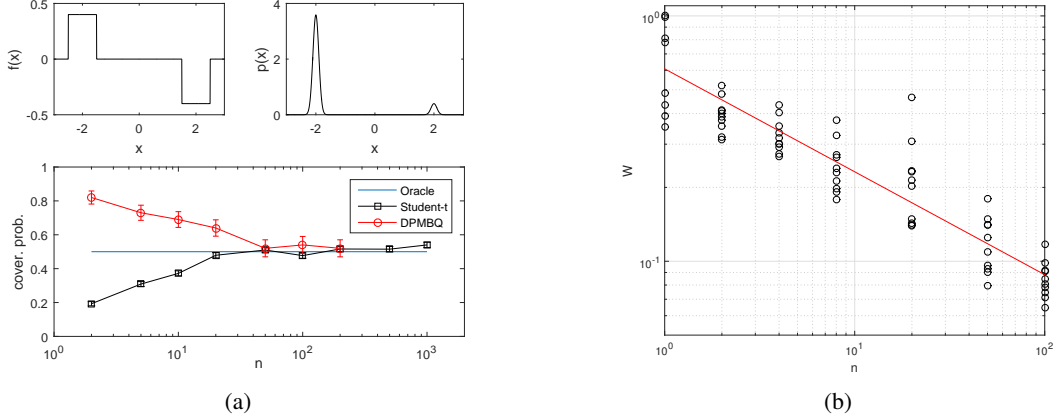

Figure 1: Simulated data results. (a) Comparison of coverage frequencies for the simulation experiments. (b) Convergence assessment: Wasserstein distance ($W$) between the posterior in Eqn. 11 and the true value of the integral, is presented as a function of the number $n$ of data points. [Circles represent independent realisations and the linear trend is shown in red.]

For comparison, we considered the default 50% confidence interval description of numerical error

$$\left(\bar{f} - t^* \frac{s}{\sqrt{n}}, \bar{f} + t^* \frac{s}{\sqrt{n}}\right) \qquad (12)$$

where $\bar{f} = n^{-1}\Sigma_{i=1}^n f(x_i)$, $s^2 = (n-1)^{-1}\Sigma_{i=1}^n (f(x_i) - \bar{f})^2$ and $t^*$ is the 50% level for a Student's $t$-distribution with $n-1$ degrees of freedom. It is well-known that Eqn. 12 is a poor description of numerical error when $n$ is small [c.f. "Monte Carlo is fundamentally unsound" 27]. For example, with $n = 2$, in the extreme case where, due to chance, $f(x_1) \approx f(x_2)$, it follows that $s \approx 0$ and no numerical error is acknowledged. This fundamental problem is *resolved through the use of prior information* on the form of both $f$ and $p$ in DPMBQ. The appropriateness of DPMBQ therefore depends crucially on the prior. The proposed method is further distinguished from Eqn. 12 in that the distribution over numerical error is fully non-parametric, not e.g. constrained to be Student-$t$.

**Empirical Results** Coverage frequencies are shown in Fig. 1a for a specific integration task $(f_0, p_0)$, that was deliberately selected to be difficult for Eqn. 12 due to the rare event represented by the mass at $x = 2$. These were compared against central 50% posterior credible intervals produced under DPMBQ. These are the frequency with which the confidence/credible interval contain the true value of the integral, here estimated with 100 independent realisations for DPMBQ and 1000 for the (less computational) standard method (standard errors are shown for both). Whilst it offers correct coverage in the asymptotic limit, Eqn. 12 can be seen to be over-confident when $n$ is small, with coverage often less than 50%. In contrast, DPMBQ accounts for the fact $p$ is being estimated and provides conservative estimation about the extent of numerical error when $n$ is small.

To present results that do not depend on a fixed coverage level (e.g. 50%), we next measured convergence in the Wasserstein distance $W = \int |p(f) - p_0(f_0)| \, d[p(f) \mid X, f(X)]$. In particular we explored whether the theoretical rate of $n^{-1/4+\epsilon}$ was realised. (Note that the theoretical result applied just to fixed hyper-parameters, whereas the experimental results reported involved hyper-parameters that were marginalised, so that this is a non-trivial experiment.) Results in Fig. 1b demonstrated that $W$ scaled with $n$ at a rate which was consistent with the theoretical rate claimed. Full experimental results on our polynomial test bed, reported in detail in the Supplemental Information, revealed that $W$ was larger for higher-degree polynomials (i.e. more complex integrands $f$), while $W$ was insensitive to the number of mixture components (i.e. to more complex distributions $p$). The latter observation may be explained by the fact that the kernel mean $\mu$ is a smoothed version of the distribution $p$ and so is not expected to be acutely sensitive to variation in $p$ itself.

## 3.2 Application to a Computational Cardiac Model

**The Model** The computation model considered in this paper is due to [24] and describes the mechanics of the left and right ventricles through a heart beat. In brief, the model geometry (Fig. 2a,

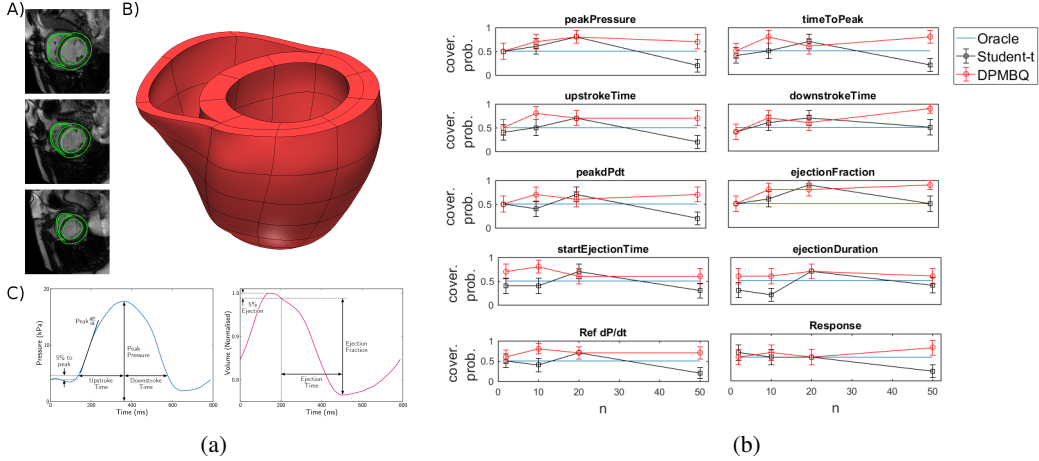

(a)                                                                          (b)

Figure 2: Cardiac model results: (a) Computational cardiac model. A) Segmentation of the cardiac MRI. B) Computational model of the left and right ventricles. C) Schematic image showing the features of pressure (left) and volume transient (right). (b) Comparison of coverage frequencies, for each of 10 numerical integration tasks defined by functionals $g_j$ of the cardiac model output.

top right) is described by fitting a $C^1$ continuous cubic Hermite finite element mesh to segmented magnetic resonance images (MRI; Fig. 2a, top left). Cardiac electrophysiology is modelled separately by the solution of the mono-domain equations and provides a field of activation times across the heart. The passive material properties and afterload of the heart are described, respectively, by a transversely isotropic material law and a three element Windkessel model. Active contraction is simulated using a phenomenological cellular model, with spatial variation arising from the local electrical activation times. The active contraction model is defined by five input parameters: $t_r$ and $t_d$ are the respective constants for the rise and decay times, $T_0$ is the reference tension, $a_4$ and $a_6$ respectively govern the length dependence of tension rise time and peak tension. These five parameters were concatenated into a vector $x \in \mathbb{R}^5$ and constitute the model inputs. The model is fitted based on training data $y$ that consist of functionals $g_j : \mathbb{R}^5 \to \mathbb{R}$, $j = 1, \ldots, 10$, of the pressure and volume transient morphology during baseline activation and when the heart is paced from two leads implanted in the right ventricle apex and the left ventricle lateral wall. These 10 functionals are defined in the Supplemental Information; a schematic of the model and fitted measurements are shown in Fig. 2a (bottom panel).

**Test Functions** The distribution $p(\mathrm{d}x)$ was taken to be the posterior distribution over model inputs $x$ that results from an improper flat prior on $x$ and a squared-error likelihood function: $\log p(x) = \text{const.} + \frac{1}{0.1^2} \sum_{j=1}^{10}(y_j - g_j(x))^2$. The training data $y = (y_1, \ldots, y_{10})$ were obtained from clinical experiment. The task we considered is to compute posterior expectations for functionals $f(x)$ of the model output produced when the model input $x$ is distributed according to $p(\mathrm{d}x)$. This represents the situation where a fitted model is used to predict response to a causal intervention, representing a clinical treatment. For assessment of the DPMBQ method, which is our principle aim in this experiment, we simply took the test functions $f$ to be each of the physically relevant model outputs $g_j$ in turn (corresponding to no causal intervention). This defined 10 separate numerical integration problems as a test bed. Benchmark values for $p_0(g_j)$ were obtained, as described in the Supplemental Information, at a total cost of $\approx 10^5$ CPU hours, which would not be routinely practical.

**Empirical Results** For each of the 10 numerical integration problems in the test bed, we computed coverage probabilities, estimated with 100 independent realisations (standard errors are shown), in line with those discussed for simulation experiments. These are shown in Fig. 2b, where we compared Eqn. 12 with central 50% posterior credible intervals produced under DPMBQ. It is seen that Eqn. 12 is usually reliable but *can* sometimes be over-confident, with coverage probabilities less than 50%. This over-confidence can lead to spurious conclusions on the predictive performance of the computational model. In contrast, DPMBQ provides a uniformly conservative quantification

of numerical error (cover. prob. $\geq 50\%$). The DPMBQ method is further distinguished from Eqn. 12 in that it entails a *joint* distribution for the 10 integrals (the unknown $p$ is shared across integrals - an instance of transfer learning across the 10 integration tasks). Fig. 2b also appears to show a correlation structure in the standard approach (black lines), but this is an artefact of the common sample set $\{x_i\}_{i=1}^n$ that was used to simultaneously estimate all 10 integrals; Eqn. 12 is still applied *independently* to each integral.

## 4 Discussion

Numerical analysis often focuses the convergence order of numerical methods, but in non-asymptotic regimes the language of probabilities can provide a richer, more intuitive and more useful description of numerical error. This paper cast the computation of integrals $p(f)$ as an estimation problem amenable to Bayesian methods [20, 9, 5]. The difficulty of this problem depends on our level of prior knowledge (rendering the problem trivial if a closed-form solution is *a priori* known) and, in the general case, on how much information we are prepared to obtain on the objects $f$ and $p$ through numerical computation [16]. In particular, we distinguish between three states of prior knowledge: (1) $f$ known, $p$ unknown, (2) $f$ unknown, $p$ known, (3) both $f$ and $p$ unknown. Case (1) is the subject of Monte Carlo methods [32] and concerns classical problems in applied probability such as estimating confidence intervals for expectations based on Markov chains. Notable recent work in this direction is [8], who obtained a point estimate $\hat{p}$ for $p$ using a kernel smoother and then, in effect, used $\hat{p}(f)$ as an estimate for the integral. The decision-theoretic risk associated with error in $\hat{p}$ was explored in [6]. Independent of integral estimation, there is a large literature on density estimation [37]. Our probabilistic approach provides a Bayesian solution to this problem, as a special case of our more general framework. Case (2) concerns functional analysis, where [26] provide an extensive overview of theoretical results on approximation of unknown functions in an information complexity framework. As a rule of thumb, estimation improves when additional smoothness can be *a priori* assumed on the value of the unknown object [see 4]. The main focus of this paper was Case (3), until now unstudied, and a transparent, general statistical method called DPMBQ was proposed.

The path-finding nature of this work raises several important questions for future theoretical and applied research. First, these methods should be extended to account for the low-rank phenomenon that is often encountered in multi-dimensional integrals [11]. Second, there is no reason, in general, to restrict attention to function values obtained at the locations in $X$. Indeed, one could first estimate $p(\mathrm{d}x)$, then select suitable locations $X'$ from at which to evaluate $f(X')$ [2]. This touches on aspects of statistical experimental design; the practitioner seeks a set $X'$ that minimises an appropriate loss functional at the level of $p(f)$; see again [6]. Third, whilst restricted to Gaussians in our experiments, further methodological work will be required to establish guidance for the choice of kernel $k$ in the GP and choice of base distribution $P_b$ in the DPM [c.f. chapter 4 of 31].

#### Acknowledgments

CJO and MG were supported by the Lloyds Register Foundation Programme on Data-Centric Engineering. SN was supported by an EPSRC Intermediate Career Fellowship. FXB was supported by the EPSRC grant [EP/L016710/1]. MG was supported by the EPSRC grants [EP/K034154/1, EP/R018413/1, EP/P020720/1, EP/L014165/1], and an EPSRC Established Career Fellowship, [EP/J016934/1]. This material was based upon work partially supported by the National Science Foundation (NSF) under Grant DMS-1127914 to the Statistical and Applied Mathematical Sciences Institute. Opinions, findings, and conclusions or recommendations expressed in this material are those of the author(s) and do not necessarily reflect the views of the NSF.

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
