[Supplementary Material]

# Supplemental Information

This supplement contains proofs, additional derivations and experimental results that complement the material in the Main Text.

## A   Proof of Theorem

Denote by $p_0$ the true distribution that gives rise to the observations in $X$. Consider inference for $p_0$ under the DPM model for $X$. Let $\mu_0(x) = p_0(k(\cdot, x)) \in \mathcal{H}$ denote the exact kernel mean. Let $\|\cdot\|_{\mathcal{H}}$ and $\langle\cdot, \cdot\rangle_{\mathcal{H}}$ denote the norm and inner product associated with $\mathcal{H}$. An important bound is derived from Cauchy-Schwarz:

$$\left| p_0(f_0) - \sum_{i=1}^n w_i f_0(x_i) \right| \leq \|f_0\|_{\mathcal{H}} \left\| \mu_0 - \sum_{i=1}^n w_i k(\cdot, x_i) \right\|_{\mathcal{H}}$$

This motivates us to study approximation of the kernel mean $\mu_0$ in a Hilbert space context. Let $\mu(x) = p(k(\cdot, x)) \in \mathcal{H}$ be the generic unknown kernel mean in the case where $p$ is an uncertain distribution. The reproducing property in $\mathcal{H}$ can be used to bound kernel mean approximation error:

$$
\begin{aligned}
\|\mu_0 - \mu\|_{\mathcal{H}}^2 &= \langle \mu_0 - \mu, \mu_0 - \mu \rangle_{\mathcal{H}} \\
&= \left\langle \int k(\cdot, x)(p_0(x) - p(x))\mathrm{d}x, \right. \\
&\qquad\qquad \left. \int k(\cdot, x')(p_0(x') - p(x'))\mathrm{d}x' \right\rangle_{\mathcal{H}} \\
&= \iint \langle k(\cdot, x), k(\cdot, x') \rangle_{\mathcal{H}} (p_0(x) - p(x))(p_0(x') - p(x')) \quad \mathrm{d}x\mathrm{d}x' \\
&\leq \sup_{x, x' \in \Omega} |k(x, x')| \times \|p_0 - p\|_1^2 \\
&\leq 4 \sup_{x, x' \in \Omega} |k(x, x')| \times d_{\mathrm{Hell}}(p_0, p)^2.
\end{aligned}
$$

The DPM model provides a posterior distribution over $p(\mathrm{d}x)$; in turn this implies a posterior distribution over the kernel mean $\mu(x)$. Denote the Hellinger distance $d_{\mathrm{Hell}}(p_0, p)$ and recall that, for two densities $p_0$, $p$, we have $\|p_0 - p\|_1 \leq 2d_{\mathrm{Hell}}(p_0, p)$. Under assumptions (A2-5) of the theorem, [8, Thm. 6.2] established that the DP location-scale mixture model satisfies $d_{\mathrm{Hell}}(p_0, p) = O_P(n^{-1/2+\epsilon})$, where $\epsilon > 0$ denotes a generic positive constant that can be arbitrarily small. Thus, in the posterior, $\|\mu_0 - \mu\|_{\mathcal{H}}^2 = O_P(n^{-1+\epsilon})$.

Let $\mu_{0,n}(\cdot) = \mu_0(X)k(X, X)^{-1}k(X, \cdot) \in \mathcal{H}$. The idealised BQ posterior, where $p(\mathrm{d}x)$ is known, takes the form

$$[p(f) \mid p_0, X, f_0(X)] = \mathrm{N}(\langle f_0, \mu_{0,n}\rangle_{\mathcal{H}}, \|\mu_0 - \mu_{0,n}\|_{\mathcal{H}}^2),$$

as shown in [3]. Let $\mu_n(\cdot) = \mu(X)k(X, X)^{-1}k(X, \cdot) \in \mathcal{H}$. For the DPMBQ posterior, where $p(\mathrm{d}x)$ is unknown, we have the conditional distribution

$$[p(f) \mid p, X, f_0(X)] = \mathrm{N}(\langle f_0, \mu_n\rangle_{\mathcal{H}}, \|\mu - \mu_n\|_{\mathcal{H}}^2).$$

Our aim is to relate the DPMBQ posterior to the idealised BQ posterior. To this end, it is claimed that:

$$\|\mu_0 - \mu_n\|_{\mathcal{H}}^2 \leq \|\mu_0 - \mu_{0,n}\|_{\mathcal{H}}^2 + n^{1/2}\|\mu_0 - \mu\|_{\mathcal{H}}^2. \tag{1}$$

Here we have decomposed the estimation error $\mu_0 - \mu_n$ into a term $\mu_0 - \mu_{0,n}$, that represents the error of the idealised BQ method, and a term $\mu_0 - \mu$ that captures the fact that the true mean element $\mu_0$ is unknown.

To prove the claim, we follow Lemma 2 in [3]: Write $\epsilon(X) = \mu(X) - \mu_0(X)$ and deduce that

$$
\begin{aligned}
&\|\mu_0 - \mu_n\|_{\mathcal{H}}^2 \\
&= \left\| \int k(x, \cdot) p_0(\mathrm{d}x) - \mu(X)^\top k(X, X)^{-1} k(X, \cdot) \right\|_{\mathcal{H}}^2 \\
&= p_0 \otimes p_0(k) - 2\mu(X)^\top k(X, X)^{-1} \mu_0(X) + \mu(X)^\top k(X, X)^{-1} \mu(X) \\
&= p_0 \otimes p_0(k) - 2(\epsilon(X) + \mu_0(X))^\top k(X, X)^{-1} \mu_0(X) \\
&\qquad + (\epsilon(X) + \mu_0(X))^\top k(X, X)^{-1} (\epsilon(X) + \mu_0(X)) \\
&= p_0 \otimes p_0(k) - 2\mu_0(X)^\top k(X, X)^{-1} \mu_0(X) + \mu_0(X)^\top k(X, X)^{-1} \mu_0(X) \\
&\qquad + \epsilon(X)^\top k(X, X)^{-1} \epsilon(X) \\
&= \|\mu_0 - \mu_{0,n}\|_{\mathcal{H}}^2 + \epsilon(X)^\top k(X, X)^{-1} \epsilon(X).
\end{aligned}
\tag{2}
$$

Let $\mathcal{H} \otimes \mathcal{H}$ denote the tensor product of Hilbert spaces [1, Sec. 1.4.6]. Then the second term in Eqn. 2 is non-negative and can be bounded using the reproducing properties of both $\mathcal{H}$ and $\mathcal{H} \otimes \mathcal{H}$:

$$
\begin{aligned}
\epsilon(X)^\top k(X, X)^{-1} \epsilon(X) &= \sum_{i,i'=1}^{n} [k(X, X)^{-1}]_{i,i'} \langle \mu - \mu_0, k(\cdot, x_i) \rangle_{\mathcal{H}} \langle \mu - \mu_0, k(\cdot, x_{i'}) \rangle_{\mathcal{H}} \\
&= \left\langle (\mu - \mu_0) \otimes (\mu - \mu_0), \sum_{i,i'=1}^{n} \begin{matrix} [k(X, X)^{-1}]_{i,i'} \\ \times k(\cdot, x_i) \otimes k(\cdot, x_{i'}) \end{matrix} \right\rangle_{\mathcal{H} \otimes \mathcal{H}} \\
&\leq \|\mu_0 - \mu\|_{\mathcal{H}}^2 \left\| \sum_{i,i'=1}^{n} [k(X, X)^{-1}]_{i,i'} k(\cdot, x_i) \otimes k(\cdot, x_{i'}) \right\|_{\mathcal{H} \otimes \mathcal{H}},
\end{aligned}
$$

where the final inequality is Cauchy-Schwarz. The latter factor evaluates to $n^{1/2}$, again using the reproducing property for $\mathcal{H} \otimes \mathcal{H}$:

$$
\begin{aligned}
&\left\| \sum_{i=1}^{n} \sum_{i'=1}^{n} [k(X, X)^{-1}]_{i,i'} k(\cdot, x_i) \otimes k(\cdot, x_{i'}) \right\|_{\mathcal{H} \otimes \mathcal{H}}^2 \\
&= \sum_{i,i',j,j'} \begin{matrix} [k(X, X)^{-1}]_{i,i'} [k(X, X)^{-1}]_{j,j'} \\ \times \langle k(\cdot, x_i) \otimes k(\cdot, x_{i'}), k(\cdot, x_j) \otimes k(\cdot, x_{j'}) \rangle_{\mathcal{H} \otimes \mathcal{H}} \end{matrix} \\
&= \sum_{i,i',j,j'} [k(X, X)^{-1}]_{i,i'} [k(X, X)^{-1}]_{j,j'} [k(X, X)]_{i,j} [k(X, X)]_{i',j'} \\
&= \operatorname{tr}[k(X, X) k(X, X)^{-1} k(X, X) k(X, X)^{-1}] \\
&= n.
\end{aligned}
$$

This establishes that the claim holds.

From Lemmas 1 and 3 in [3], we have that the idealised BQ estimate based on the bounded kernel $k$ satisfies $\|\mu_0 - \mu_{0,n}\|_{\mathcal{H}} = O_P(n^{-1/2})$. Indeed, $\|\mu_0 - \mu_{0,n}\|_{\mathcal{H}} \leq \|\mu_0 - \hat{\mu}_{0,n}\|_{\mathcal{H}}$, where

$$
\hat{\mu}_{0,n} = \frac{1}{n} \sum_{i=1}^{n} k(\cdot, x_i)
$$

is the Monte Carlo estimate for the kernel mean [Lemma 3 of 3]. As $k$ is bounded, the norm $\|\mu_0 - \hat{\mu}_{0,n}\|_{\mathcal{H}}$ vanishes as $O_P(n^{-1/2})$ [Lemma 1 of 3]. Combining the above results in Eqn. 1, we obtain

$$
\begin{aligned}
\|\mu_0 - \mu_n\|_{\mathcal{H}}^2 &= O_P(n^{-1}) + n^{1/2} \times O_P(n^{-1+\epsilon}) \\
&= O_P(n^{-1/2+\epsilon}).
\end{aligned}
$$

To finish, recall that for DPMBQ we have the random variable representation

$$p(f) = \langle f_0, \mu_n \rangle_{\mathcal{H}} + \|\mu - \mu_n\|_{\mathcal{H}} \, \xi,$$

where $\xi \sim N(0, 1)$ is independent of $X$. Thus, from the triangle inequality followed by Cauchy-Schwarz:

$$
\begin{aligned}
|p_0(f_0) - p(f)| &= |\langle f_0, \mu_0 \rangle_{\mathcal{H}} - \langle f_0, \mu_n \rangle_{\mathcal{H}} - \|\mu - \mu_n\|_{\mathcal{H}} \, \xi| \\
&\leq |\langle f_0, \mu_0 - \mu_n \rangle_{\mathcal{H}}| + \|\mu - \mu_n\|_{\mathcal{H}} \, |\xi| \\
&\leq |\langle f_0, \mu_0 - \mu_n \rangle_{\mathcal{H}}| + [\|\mu - \mu_0\|_{\mathcal{H}} + \|\mu_0 - \mu_n\|_{\mathcal{H}}] \, |\xi| \\
&\leq \|f_0\|_{\mathcal{H}} \|\mu_0 - \mu_n\|_{\mathcal{H}} + O_P(n^{-1/2+\epsilon}) + O_P(n^{-1/4+\epsilon}) \\
&= O_P(n^{-1/4+\epsilon}).
\end{aligned}
$$

Denote the DPMBQ posterior distribution with $\mathbb{P}_n = [p(f) \mid X, f(X)]$. Then for $\delta > 0$ fixed, the posterior mass $\mathbb{P}_n[(\infty, p_0(f_0) - \delta) \cup (p_0(f_0) + \delta, \infty)] = O_P(n^{-1/4+\epsilon})$. This completes the proof.

# B  Computational Details

This section describes the computation for DPMBQ. The model admits the following straight-forward sampler:

1. draw $\theta$ from the hyper-prior $[\theta]$

2. draw $\phi_{1:n}$ from $[\phi_{1:n} \mid X, \theta]$ (via a Gibbs sampler)

3. draw $p$ from $[\mathrm{d}p \mid \phi_{1:n}]$ (via stick-breaking)

4. draw $p(f)$ from $[p(f) \mid X, f(X), p, \theta]$ (via BQ)

For step (2), it is convenient (but not essential) to use a conjugate base distribution $P_b$. In the case of a Gaussian model $\psi$, the normal inverse-gamma distribution, parametrised with $\mu_0 \in \mathbb{R}$, $\lambda_0, \alpha_0, \beta_0 \in (0, \infty)$, permits closed-form conditionals and facilitates an efficient Gibbs sampler. Full details are provided in supplemental Sec. B.1. (Note that the conjugate base distribution does not fall within the scope of the theorem; however the use of a more general Metropolis-within-Gibbs scheme enables computation from such models with trivial modification.) In all experiments below we fixed hyper-parameters to default values $\lambda_0 = \alpha_0 = \beta_0 = 1$, $\mu_0 = 0$; there was no noticeable dependence of inferences on these choices, which are several levels removed from $p(f)$, the unknown of interest.

This direct scheme admits several improvements: e.g. (a) stratified or QMC sampling of $\theta$ in step (1); (b) Rao-Blackwellisation of the additional randomisation in $p(f)$, to collapse steps (3) and (4) [2]; (c) the Gibbs sampler of [5] can be replaced by more sophisticated alternatives, such as [11]. Indeed, one need not sample from the prior $[\theta]$ and instead target the hyper-parameter posterior with MCMC. In experiments, the straight-forward scheme outlined here was more than adequate to obtain samples from the DPMBQ model. Thus we implemented this basic sampler and leave the above extensions as possible future work.

## B.1  Gibbs Sampler

This section derives the conditional distributions that are needed for an efficient Gibbs sampler that targets $[\phi \mid X, \theta]$. The main result is presented in the proposition below:

**Proposition.** *Consider the multivariate Gaussian model $\psi(\mathrm{d}x; \phi) = N(\mathrm{d}x | \phi_1, \mathrm{diag}(\phi_2))$, with mean vector $\phi_1 \in \mathbb{R}^d$ and marginal variance vector $\phi_2 \in \mathbb{R}^d$. Consider the base distribution $P_b(\mathrm{d}\phi)$ composed of independent normal inverse-gamma $\mathrm{NIG}(\phi_{1,k}, \phi_{2,k} | \mu_0, \lambda_0, \alpha_0, \beta_0)$ components with $\mu_0 \in \mathbb{R}$, $\lambda_0, \alpha_0, \beta_0 \in (0, \infty)$ for $k = 1, \ldots, d$. Denote $\phi_i = (\phi_{i,1}, \phi_{i,2})$ and $\phi_{(-i)} = (\phi_1, \ldots, \phi_{i-1}, \phi_{i+1}, \ldots, \phi_n)$. For this conjugate choice, we have the closed-form posterior conditional*

$$[\phi_i \mid \phi_{(-i)}, X, \theta] = \omega_0 Q_i + \sum_{j \neq i} \omega_j \delta_{\phi_j}$$

where $Q_i$ is composed of independent $\text{NIG}(\phi_{i,1,k}, \phi_{i,2,k} | \mu_{i,k}, \lambda_{i,k}, \alpha_{i,k}, \beta_{i,k})$ components and

$$\begin{bmatrix} \omega_0 \\ \omega_j \end{bmatrix} \quad \propto \quad \begin{bmatrix} \alpha \prod_{k=1}^{d} \frac{1}{2\pi^{1/2}} \frac{\lambda_0^{1/2}}{\lambda_{i,k}^{1/2}} \frac{\beta_0^{\alpha_0}}{\beta_{i,k}^{\alpha_{i,k}}} \frac{\Gamma(\alpha_{i,k})}{\Gamma(\alpha_0)} \\ \text{N}(x_i|\phi_{j,1}, \text{diag}(\phi_{j,2})) \end{bmatrix}$$

$$\mu_{i,k} = \frac{\lambda_0 \mu_0 + x_{i,k}}{\lambda_0 + 1}$$

$$\lambda_{i,k} = \lambda_0 + 1$$

$$\alpha_{i,k} = \alpha_0 + \frac{1}{2}$$

$$\beta_{i,k} = \beta_0 + \frac{1}{2}(\lambda_0 \mu_0^2 + x_{i,k}^2 - \lambda_{i,k}\mu_{i,k}^2).$$

*Proof.* From Theorem 1 of [6], also known as "Bayes' theorem for DPs", we have that the prior $P \sim \text{DP}(\alpha, P_b)$ and the likelihood $\phi_i \sim P$ (independent) lead to a posterior

$$P \mid \phi_{(-i)} \sim \text{DP}\left(\alpha + n - 1, \frac{1}{\alpha + n - 1}\left(\alpha P_b + \sum_{j \neq i} \delta_{\phi_j}\right)\right).$$

It follows that, for a measurable set $A$,

$$\begin{aligned} \text{Prob}[\phi_i \in A \mid \phi_{(-i)}] &= \mathbb{E}[P(A) \mid \phi_{(-i)}] \\ &= \frac{1}{\alpha + n - 1}\left(\alpha P_b(A) + \sum_{j \neq i} \delta_{\phi_j}(A)\right). \end{aligned}$$

From (standard) Bayes' theorem,

$$\begin{aligned} [\phi_i \mid \phi_{(-i)}] &= \frac{[X \mid \phi_{1:n}][\phi_i \mid \phi_{(-i)}]}{[X \mid \phi_{(-i)}]} \\ &\propto [X \mid \phi_{1:n}][\phi_i \mid \phi_{(-i)}] \propto [x_i \mid \phi_i][\phi_i \mid \phi_{(-i)}] \end{aligned}$$

and combining the two above results, in the case of a Gaussian model $\psi(dx_i; \phi_i)$ with mean vector $\phi_{i,1}$ and marginal variance vector $\phi_{i,2}$, leads to

$$\begin{aligned} [\phi_i \mid \phi_{(-i)}] &\propto \text{N}(x_i|\phi_{i,1}, \text{diag}(\phi_{i,2})) \times \left(\alpha P_b(\phi_i) + \sum_{j \neq i} \delta_{\phi_j}(\phi_i)\right) \\ &= \alpha \text{N}(x_i|\phi_{i,1}, \text{diag}(\phi_{i,2}))P_b(\phi_i) + \sum_{j \neq i} \text{N}(x_i|\phi_{j,1}, \text{diag}(\phi_{j,2}))\delta_{\phi_j}(\phi_i), \end{aligned}$$

where $\phi_i = (\phi_{i,1}, \phi_{i,2})$ with $\phi_{i,1} \in \mathbb{R}^d$ and $\phi_{i,2} \in (0, \infty)^d$.

For closed-form expressions, $P_b$ must be taken conjugate to the Gaussian model:

$$\begin{aligned} P_b(\phi_i) &= \prod_{k=1}^{d} \text{NIG}(\phi_{i,1,k}, \phi_{i,2,k} | \mu_0, \lambda_0, \alpha_0, \beta_0) \\ &= \prod_{k=1}^{d} \text{N}(\phi_{i,1,k} | \mu_0, \lambda_0^{-1}\phi_{i,2,k})\text{IG}(\phi_{i,2,k} | \alpha_0, \beta_0), \end{aligned}$$

in the obvious notation $\phi_{i,j} = (\phi_{i,j,1}, \ldots, \phi_{i,j,d})$. Thus

$$\begin{aligned} \text{N}(x_i|\phi_{i,1}, \text{diag}(\phi_{i,2}))P_b(\phi_i) &= \text{N}(x_i|\phi_{i,1}, \text{diag}(\phi_{i,2})) \\ &\quad \times \prod_{k=1}^{d} \text{NIG}(\phi_{i,1,k}, \phi_{i,2,k} | \mu_0, \lambda_0, \alpha_0, \beta_0) \\ &= \omega_0 \times \prod_{k=1}^{d} \text{NIG}(\phi_{i,1,k}, \phi_{i,2,k} | \mu_{i,k}, \lambda_{i,k}, \alpha_{i,k}, \beta_{i,k}), \end{aligned}$$

where

$$\omega_0 = \prod_{k=1}^{d} \frac{1}{2\pi^{1/2}} \frac{\lambda_0^{1/2}}{\lambda_{i,k}^{1/2}} \frac{\beta_0^{\alpha_0}}{\beta_{i,k}^{\alpha_{i,k}}} \frac{\Gamma(\alpha_{i,j})}{\Gamma(\alpha_0)}$$

$$\mu_{i,k} = \frac{\lambda_0 \mu_0 + x_{i,k}}{\lambda_0 + 1}$$

$$\lambda_{i,k} = \lambda_0 + 1$$

$$\alpha_{i,k} = \alpha_0 + \frac{1}{2}$$

$$\beta_{i,k} = \beta_0 + \frac{1}{2}(\lambda_0 \mu_0^2 + x_{i,k}^2 - \lambda_{i,k} \mu_{i,k}^2).$$

This completes the proof. □

In all experiments the Gibbs sampler was initialised at $\phi_{i,1,k} = x_{i,k}$ and $\phi_{i,2,k} = 1$ and run until a convergence criteria was satisfied. In this way we produced samples from $[\phi_{1:n} \mid X, \theta]$ for the direct sampling scheme outlined in the main text.

## B.2 Tensor Structure for Multi-Dimensional Integrals

This section describes how multi-dimensional integration problems on a tensor-structured domain $\Omega = \Omega_1 \otimes \cdots \otimes \Omega_d$ can be decomposed into a tensor product of univariate integration problems. This construction was used to produce the results in the Main Text, as well as in Sec. C.2 of the Supplement.

Assume a tensor product kernel

$$k(x, x') = k_1(x_1, x_1') \times \cdots \times k_d(x_d, x_d')$$

on $\Omega \times \Omega$, together with a product model

$$\psi(\mathrm{d}x; \phi) = \psi_1(\mathrm{d}x_1; \phi_1) \times \cdots \times \psi_d(\mathrm{d}x_d; \phi_d).$$

Then a generic draw from $[p \mid \phi_{1:n}]$ has the form

$$p(\mathrm{d}x) = \sum_{j=1}^{\infty} w_j \psi_1(\mathrm{d}x_1; \varphi_{j,1}) \times \cdots \times \psi_d(\mathrm{d}x_d; \varphi_{j,d}),$$

where $\varphi_j \sim P$ are independent with $\varphi_j = (\varphi_{j,1}, \ldots, \varphi_{j,d})$, and the corresponding kernel mean is

$$\mu(x) = \sum_{j=1}^{\infty} w_j \prod_{i=1}^{d} \left( \int_{\Omega_i} k_i(x_i, x_i') \psi_i(x_i'; \varphi_{j,i}) \mathrm{d}x_i' \right).$$

The initial error $p \otimes p(k)$ is derived as

$$p \otimes p(k) = \sum_{j,j'=1}^{\infty} w_j w_{j'} \prod_{i=1}^{d} \int_{\Omega_i} k_i(x_i, x_i') \psi_i(x_i; \varphi_{j,i}) \psi_i(x_i'; \varphi_{j,i}) \mathrm{d}x_i \mathrm{d}x_i'.$$

For an efficient Gibbs sampler, as in Sec. B.1, the prior model on the mixing distribution $P(\mathrm{d}\phi)$ was taken as a tensor product of $\mathrm{DP}(\alpha, P_{b,i})$ priors where $P_{b,i}(\mathrm{d}x_i)$ is a base distribution on $\Omega_i$. The experiments of Sec. C.2 were performed as explained above, where the individual components $k_i$, $\psi_i$ and $P_{b,i}$ were taken to be the same as used for the simulation examples in Sec. C.

## C Experimental Set-Up and Results

Two simulation studies were undertaken, based on polynomial test functions where the true integral is known in closed-form (Sec. C.1) and based on differential equations where the true integral must be estimated with brute-force computation (Sec. C.2).

## C.1 Flexible Polynomial Test Bed

To assess the performance of the DPMBQ method, we considered independent data $x_1, \ldots, x_n$ generated from a known distribution $p(\mathrm{d}x)$. In addition, the function $f(x)$ was fixed and known, so that overall the exact value of the integral $p(f)$ provided a known benchmark.

For illustration, we focused on the generic class of one-dimensional test problems obtained when $p(\mathrm{d}x)$ is a Gaussian mixture distribution

$$p(\mathrm{d}x) = \sum_{i=1}^{m} r_i \mathrm{N}(\mathrm{d}x; c_i, s_i^2)$$

defined on $\Omega = \mathbb{R}$, where $c_i \in \mathbb{R}$, $r_i, s_i \in [0, \infty)$, $\sum_{i=1}^{m} r_i = 1$, and the function $f(x)$ is a polynomial

$$f(x) = \sum_{i=1}^{q} a_i x^{b_i}$$

where $a_i \in \mathbb{R}$ and $b_i \in \mathbb{N}_0$. For this problem class, the integral $p(f)$ is computable in closed-form and the generic approximation properties of Gaussian mixtures and polynomials provide an expressive test-bed. In addition, the GP prior with mean function $m_\theta(x) = 0$ and Gaussian covariance function

$$k_\theta(x, x') = \zeta \exp(-(x - x')^2 / 2\lambda^2)$$

was employed with $\zeta = 1$ fixed. This choice provides a closed-form kernel mean for assessment purposes, with standard Gaussian calculations analogous to those performed in the Main Text.

**Illustration**   Consider the toy problem where $f(x) = 1 + x - 0.1x^3$, $p(\mathrm{d}x) = \mathrm{N}(\mathrm{d}x; 0, 1)$, such that the true integral $p(f) = 1$ is known in closed-form. For the kernel $k_\theta$ we initially fixed the hyper-parameter $\lambda$ at a default value $\lambda = 1$. The concentration hyper-parameter $\alpha$ was initially fixed to $\alpha = 1$ (the *unit information* DP prior). For all experiments, the stick breaking construction described in the Main Text was truncated after the first $N = 500$ terms; at this level results were invariant to further increases in $N$. In Fig. 1 we present realisations of the posterior distributions $[\mu \mid X]$ and $[p(f) \mid X, f(X)]$ at two sample sizes, (a) $n = 10$ and (b) $n = 100$. In this case each posterior contains the true value $p_0(f_0)$ of the integral in its effective support region. The posterior variance is greatly inflated with respect to the idealised case in which $p(\mathrm{d}x)$, and hence the kernel mean $\mu$, is known. This is intuitively correct and reflects the increased difficulty of the problem in which both $f(x)$ and $p(\mathrm{d}x)$ are *a priori* unknown.

**Detailed Results**   To explore estimator convergence in detail, we considered the general simulation set-up above and measured estimator performance with the Wasserstein (or *earth movers'*) distance:

$$W = \int |p(f) - p_0(f_0)| \, \mathrm{d}[p(f) \mid X, f(X)].$$

Consistent estimation, as defined in the Main Text, is implied by convergence in Wasserstein distance. It should be noted that consistent estimation does not imply correct coverage of posterior credible intervals [7]; this aspect is left for future work.

There are three main questions that we address below; these concern dependence of the approximation properties of the posterior $[p(f) \mid X, f(X)]$ on (i) the number $n$ of data, (ii) the complexity of the distribution $p(\mathrm{d}x)$, and (iii) the complexity of the function $f(x)$. Our results can be summarised as follows:

- **Effect of the number $n$ of data:** As $n$ increases, we expect contraction of the posterior measure over $[\mu \mid X]$ onto the true kernel mean. Hence, in the limit of infinite data, the resultant integral estimates will coincide with those of BQ. However, the rate of convergence of the proposed method could be much slower compared to the idealised case in which $p(\mathrm{d}x)$, and hence $\mu(x)$, is *a priori* known.

  The problem of Fig. 1 was considered in a more general setting where the hyper-parameters $\theta$ are assigned prior distributions and are subsequently marginalised out. For these results, the kernel parameter $\lambda$ was assigned a $\mathrm{Gam}(2, 1)$ hyper-prior and the concentration parameter $\alpha$ was assigned a $\mathrm{Exp}(1)$ hyper-prior; these were employed for the remainder.

Figure 1: Illustration; computation of $p(f)$ where both $f(x)$ and $p(\mathrm{d}x)$ are *a priori* unknown. Partial information on $p(\mathrm{d}x)$ is provided as $n$ draws $x_i \sim p(\mathrm{d}x)$. Partial information on $f(x)$ is provided by the values $f(x_i)$ at each of the $n$ locations. Left: Bayesian estimation of the kernel mean $\mu$, that characterises the unknown distribution $p(\mathrm{d}x)$. Right: Posterior distribution over the value of the integral $p(f)$ (dashed line); for reference, the truth (red line) and the posterior that would be obtained *if* $p(\mathrm{d}x)$ was known (dotted line) are also shown. Two sample sizes, (top) $n = 10$, (bottom) $n = 100$, are presented.

Results in Fig. 3 showed that the posterior $[p(f) \mid X, f(X)]$ appears to converge to the true value of the integrand (in the Wasserstein sense) as the number $n$ of data are increased. The slope of the trend line was $\approx -1/4$, in close agreement with the theoretical analysis. This does not resemble the rapid posterior contraction results established in BQ when $p(\mathrm{d}x)$ is *a priori* known, which can be exponential for the Gaussian kernel [3]. This reflects the more challenging nature of the estimation problem when $p(\mathrm{d}x)$ is unavailable in closed-form.

- **Effect of the complexity of** $p(\mathrm{d}x)$**:** It is anticipated that a more challenging inference problem for $p(\mathrm{d}x)$ entails poorer estimation performance for $p(f)$. To investigate, the complexity of $p(\mathrm{d}x)$ was measured as the number $m$ of mixture components. For this experiment, the number $m$ of mixture components was fixed, with weights $(r_1, \ldots, r_m)$ drawn from $\mathrm{Dir}(2)$. The location parameters $c_i$ were independent draws from $N(0, 1)$ and the scale parameters $s_i$ were independent draws from $\mathrm{Exp}(1)$.

Results in Fig. 2 (left), which were based on $n = 20$, did not demonstrate a clear effect. This was interesting and can perhaps be explained by the fact that $\mu(x)$ is a kernel-smoothed version of $p(\mathrm{d}x)$ and thus is somewhat robust to fluctuations in $p(\mathrm{d}x)$.

Figure 2: Empirical investigation. The Wasserstein distance, $W$, between the posterior $[p(f) \mid X, f(X)]$ and the true value of the integral is presented as a function of (left) the number $m$ of mixture components that constitute $p(\mathrm{d}x)$, and (right) the degree $q$ of the polynomial function $f(x)$ whose integral is to be determined. [Circles represent independent realisations of $W$, while in (right) a linear trend line (red) is shown.]

Figure 3: Empirical investigation. The example of Fig. 1 was again considered, this time marginalising over hyper-parameters $\lambda$ and $\alpha$. The Wasserstein distance, $W$, between the posterior $[p(f) \mid X, f(X)]$ and the true value of the integral, is presented as a function of the number $n$ of data points. [Circles represent independent realisations, while a linear trend line (red) is shown.]

- **Effect of the complexity of $f(x)$:** A more challenging inference problem for $f(x)$ ought to also entails poorer estimation. To investigate, the complexity of $f(x)$ was measured as the degree $q$ of this polynomial. For each experiment, $q$ was fixed and the coefficients $a_i$ were independent draws from $\mathrm{N}(0,1)$.

  Results in Fig. 2 (right), based on $n = 20$, showed that the posterior was more accurate for larger $q$, this time in agreement with intuition.

## C.2  Goodwin Oscillator

Our second simulation experiment considered the computation of Bayesian forecasts based on a 5-dimensional computer model.

For a manageable benchmark we took a computer model that is well-understood; the *Goodwin oscillator*, which is prototypical for larger models of complex chemical systems [9]. The oscillator considers a competitive molecular dynamic, expressed as a system of ordinary differential equations

Figure 4: Application to Bayesian forecasting. Left: Data on two species, $S_1$ and $S_2$, generated from the Goodwin oscillator, a system of differential equations that contain five unknown parameters. The forecast $p(f)$ under consideration is the posterior expected concentration of species $S_1$ at the later time point $t = 50$. Right: The Wasserstein distance, $W$, from the proposed posterior $[p(f) \mid X, f(X)]$ to the true integral is shown. Here $n$ represents the number of samples $x_i$ that were obtained from the posterior $[x \mid y]$ over the unknown parameters.

(ODEs), that induces oscillation between the concentration $z_i(t; x)$ of two species $S_i$ ($i = 1, 2$). Parameters, denoted $x$ and *a priori* unknown, included two synthesis rate constants, two degradation rate constants and one exponent parameter. Full details, that include the prior distributions over parameters used in the experiment below, can be found in [12]. From an experimental perspective, we suppose that concentrations of both species are observed at 41 discrete time points $t_j$ with uniform spacing in $[0, 40]$. Observation occurred through an independent Gaussian noise process $y_{i,j} = z_i(t_j; x) + \epsilon_{i,j}$ where $\epsilon_{i,j} \sim N(0, 0.1^2)$. Data-generating parameters were identical to [12] with model dimension $g = 3$. Fig. 4 (left) shows the full data $y = (y_{i,j})$.

The forecast that we consider here is for the concentration of $S_1$ at the later time $t = 50$. In particular we defined $f(x)$ to be equal to $z_1(50)$ and obtained $n$ samples $x_i$ from the posterior $[x \mid y]$ using tempered population Markov chain Monte Carlo (MCMC), in all aspects identical to [12]. Then, $f(x_i)$ was evaluated and stored for each $x_i$; the locations $X = (x_i)$ and function evaluations $f(X)$ are the starting point for the DPMBQ method.

This prototypical model is small enough for numerical error to be driven to zero via repeated numerical simulation of the ODEs, providing us with a benchmark. Nevertheless, the key features that motivate our work are present here: (i) The forecast function $f(x)$ is expensive and black-box, being a long-range solution of a system of ODEs and requiring that the global solution error is carefully controlled. (ii) The task of obtaining samples $x_i$ is costly, as each evaluation of the likelihood $[y \mid x]$, and hence the posterior $[x \mid y]$, requires the solution of a system of ODEs.

Performance was examined through the Wasserstein distance to the true forecast $p_0(f_0)$, the latter obtained through brute-force simulation. The multi-dimensional integral was modelled as a tensor product of one-dimensional integrals, as described in Sec. B.2 in the supplement. This allowed the uni-variate model from Sec. C to be re-used at minimal effort. Results, in Fig. 4 (right), indicated that the posterior was consistent. Note that the Wasserstein distances are large for this problem, reflecting the greater uncertainties that are associated with a 5-dimensional integration problem with only $n < 10^2$ draws from $p(\mathrm{d}x)$.

An extension of this framework, not considered here, would use a probabilistic ODE solver in tandem with DPMBQ to model the approximate nature of numerical solution to the ODEs in the reported forecasts [13, 10].

### C.3 Cardiac Model Experiment

**Test Functionals $g_j$ Used in the Cardiac Model Experiment**    The 10 functionals $g_j$, that are the basis for clinical data on the cardiac model in the main text, are defined in the next paragraph:

The left ventricle pressure curve during baseline activation is characterised by the peak value (Peak Pressure), the time of the peak value (Time to Peak) and the time for pressure to rise (Upstroke Time) from 5% of the pressure change to the peak value and then fall back down (Down Stroke Time). The volume transient is described by the ratio of the left ventricle volume of blood ejected over the maximal left ventricle volume (Ejection Fraction), the time that the ventricle volume has decreased by 5% of the maximal volume (Start Ejection Time) and the time taken between the start of ejection and the point where the heart reaches its smallest left ventricle volume (Ejection Duration). The effect of pacing the heart is measured by the percentage change in the maximum rate of pressure development at baseline (Ref dPdt) and during pacing (Peak dPdt), defined as the acute haemodynamic response (Response).

**Brute-Force Computation for a Benchmark**    The samples $\{x_i\}_{i=1}^n$ from $p(\mathrm{d}x)$ can in principle be obtained via any sophisticated Markov chain Monte Carlo (MCMC) methods, such as [14, 4]. Recall that each evaluation of $p(\mathrm{d}x)$ requires $\approx 10^3$ hours, so that the MCMC method must be efficient. To reduce the computational overhead required for this project, we circumvented MCMC and instead exploited an existing, detailed empirical approximation to $p(\mathrm{d}x)$ that had been pre-computed by a subset of the authors. This consisted of a collection of $m \approx 10^3$ weighted states $(x_i, p_i)$, where the $x_i$ were selected via an *ad-hoc* adaptive Latin hypercube method, and such that the weights $p_i \propto p(x_i)$. Then, in this work, an (approximate) sample of size $n \ll m$ was obtained by sampling with replacement from the empirical distribution defined by this weighted point set. For our assessment of DPMBQ, benchmark values for each integral were computed as $\Sigma_{i=1}^m p_i f(x_i)$ for $m \approx 10^3$; note that this required a total of $\approx 10^5$ CPU hours and would not be routinely practical.