[Reviews · NeurIPS 2017]

Reviewer 1



This paper develops DPMBQ, that casts the computation of integrals p(f) as an estimation problem using a nonparametric Bayesian method, namely Dirichlet process mixtures (DPM). The authors also establish theoretical properties of the proposed method. Considering the numerical approximation of integral, DPMBQ is able to accurately estimate it when both the functional of the output from a computer model and the distribution of the inputs x of the model are unknown. Overall, the paper is clearly written and delivered. There are some concerns below. 1. Novelty. This paper develops DPMBQ by considering a DPM prior on the inputs x while assuming the Gaussian process prior on p(f). This setup allows an closed form on mu when choosing the conjugate prior. The idea is similar to the recent JASA paper "Bayesian Nonparametric Longitudinal Data Analysis" by Quintana, Johnson, Waetjen, and Gold, who proposes a Gaussian process with DPM prior on the parameters in GP. Therefore, the key idea in this paper is not new to Bayesian nonparametric community. However, I am not an expert in computer model. Can the authors explain more on the novelty of the proposed DPMBQ? 2. The combination of GP and DPM obviously make the computational burden very heavy, when n is large. Some discussion on how to incorporate some sparse GP methods into DMPBQ for computational efficiency will be useful. 3. About the theoretical analysis. The results are straightforward to derive from the consistency results of the GP due to the nice property of the DPM. About condition (4), is it easy to generalize to unbounded sigma? 4. The good results in the simulation study and real data analysis are no surprise since GP and DPM are two powerful nonparametric models that essentially approximates any continuous distributions. Therefore, of course DPMBQ will outperform the t-distribution assumption. It will be interesting to see the comparison between DPMBQ and some other methods that are not too naive.

Reviewer 2



The paper is well written and motivated. The problem is that of computing integrals involving functionals of outputs from expensive models where large number of samples cannot be readily obtained. The assessment of numerical error is therefore of great importance in some potential applications. The authors in particular address an extension of Bayesian Quadrature to address a restriction in the classic formulation in which a closed form expression is available for the distribution over inputs. p(dx) The authors extend the framework to the case where this distribution can only be sampled from. They model the unknown distribution using a Dirichlet Process Mixture and the formulation seems convincing although it must be accepted that correlated x's must be ignored in this setup. The overall methodology is sound. As neither probabilistic numerics or inference with expensive computational models are my areas of expertise, I cannot assess the positioning of this work relative to other related works in the area. However, as a general reader, the simulated numerical experiments and application to cardiac modelling were interesting and in line with the justification of purpose given in the Introduction. It would be a paper/poster that would interest myself at NIPS. However, I could equally understand that the problem is too domain-specific.

Reviewer 3



Summary The paper presents a method for assessing the uncertainty in the evaluation of an expectation over the output of a complex simulation model given uncertainty in the model parameters. Such simulation models take a long time to solve, given a set of parameters, so the task of averaging over the outputs of the simulation given uncertainty in the parameters is challenging. One cannot simply run the model so many times that error in the estimate of the integral is controlled. The authors approach the problem as an inference task. Given samples from the parameter posterior one must infer the posterior over the integral of interest. The parameter posterior can be modelled as a DP mixture and my matching the base measure of the DP mixture with the kernel used in the Gaussian process based Bayesian Quadrature (BQ) procedure for solving the integral. The method is benchmarked using synthetic and a real-world cardiac model against simply using the mean and standard error of simulations from the model for parameter samples. It is shown to be more conservative - the simple standard error approach is over-confident in the case of small sample size. Comments: This is an interesting topic and extends recent developments in probabilistic numerics to the problem of estimating averages over simulations from a model with uncertain parameters. I'm not aware of other principled approaches to solving this problem and I think it is of interest to people at NIPS. The paper is well written and easy to follow. A lot of technical detail is in the supplementary so really this is a pretty substantial paper if one considers it all together. Putting technical details in the supplement improves the readability. There is a proof of consistency of the approach in a restricted setting which gives some rigorous foundations to the approach. The matching of the mixture model components and kernel into conjugate pairs (e.g. Gaussian mixture with squared exponential kernel) seems like a neat trick to make things tractable. It is presented as an advantage that the method is conservative, e.g. the broad posterior over the quantity of interest is more likely to contain the ground truth. However, I was not sure what was the reason for the conservative behaviour and how general that behaviour is - is it generally the case that the method will be conservative or could there be cases where the procedure also underestimates the uncertainties of interest? Is this simply an empirical observation based on a restricted set of examples? I guess this is difficult to address theoretically as it is not an asymptotic issue.